# Optimized Fast Filtration-Based Sampling and Extraction Enables Precise and Absolute Quantification of the *Escherichia coli* Central Carbon Metabolome

**DOI:** 10.3390/metabo13020150

**Published:** 2023-01-18

**Authors:** Lilja Brekke Thorfinnsdottir, Laura García-Calvo, Gaute Hovde Bø, Per Bruheim, Lisa Marie Røst

**Affiliations:** Department of Biotechnology and Food Science, NTNU Norwegian University of Science and Technology, N-7491 Trondheim, Norway

**Keywords:** *Escherichia coli*, metabolite extraction, fast filtration sampling, targeted metabolite profiling, mass spectrometry, central carbon metabolism, absolute metabolite quantification, intracellular metabolite concentrations, adenylate energy charge

## Abstract

Precise and accurate quantification is a prerequisite for interpretation of targeted metabolomics data, but this task is challenged by the inherent instability of the analytes. The sampling, quenching, extraction, and sample purification conditions required to recover and stabilize metabolites in representative extracts have also been proven highly dependent on species-specific properties. For *Escherichia coli*, unspecific leakage has been demonstrated for conventional microbial metabolomics sampling protocols. We herein present a fast filtration-based sampling protocol for this widely applied model organism, focusing on pitfalls such as inefficient filtration, selective loss of biomass, matrix contamination, and membrane permeabilization and leakage. We evaluate the effect of and need for removal of extracellular components and demonstrate how residual salts can challenge analytical accuracy of hyphenated mass spectrometric analyses, even when sophisticated correction strategies are applied. Laborious extraction procedures are bypassed by direct extraction in cold acetonitrile:water:methanol (3:5:2, *v/v*%), ensuring compatibility with sample concentration and thus, any downstream analysis. By applying this protocol, we achieve and demonstrate high precision and low metabolite turnover, and, followingly, minimal perturbation of the inherent metabolic state. This allows us to herein report absolute intracellular concentrations in *E. coli* and explore its central carbon metabolome at several commonly applied cultivation conditions.

## 1. Introduction

Today the field of metabolomics is dominated by hyphenated mass spectrometry (MS) techniques offering the resolution, linear dynamic range, and sensitivity to separate, cover, and detect intracellular metabolite concentrations [1] spanning several orders of magnitude [2,3,4,5]. Metabolomics analyses are, however, challenged by the nature of the metabolome itself. Metabolites are structurally diverse, implying a broad range of physicochemical properties, have turnover rates on the scale of seconds [2,6], and are susceptible to both non-enzymatic degradation and inter-conversions [7,8], leaving them inherently unstable. This calls for focus on reproducibility, stabilization, and recovery in every step of a metabolomics workflow, including the experimental design, as both cultivation conditions and the nutrient environment influence metabolism [9,10,11]. A metabolomics sampling and extraction protocol should impose immediate metabolic arrest while minimizing perturbations to the system [12,13]. This implies strategies to separate cells from the surrounding culture medium when needed, rapidly quench enzymatic activity, perform complete extraction from the intracellular compartment (s), remove components that might else compromise accuracy, and concentrate the sample while simultaneously recovering high-turnover metabolites. The choice of strategy depends both on the properties of the model species [14,15] and the culture conditions, on the physicochemical properties of metabolites of interest [16], and on properties of the analytical methodology available for downstream analysis. Several protocols have been proposed for microbial metabolomics, relying either on concentration by filtration, cold centrifugation, or the sampling of whole culture broth. The literature is, however, contradictory with respect to the reliability of the different methods (reviewed in [17]), leaving the development of optimal sampling, quenching, extraction, and sample purification protocols an ongoing and active field of research, as summarized in [18]. 

Sampling by “cold methanol quenching”: the spraying of culture broth into cold (−40 °C) aqueous methanol (60%, *v/v*%) and subsequent removal of extracellular components by cold centrifugation, has been widespread for metabolomics studies of yeast [19,20,21]. From a rapid-quenching perspective this protocol is favored over filtration-based protocols in which quenching occurs after the separation of biomass and culture medium. Cold methanol quenching has, however, been reported to compromise cell-membrane integrity and cause extensive metabolite leakage in both yeast and bacteria [2,22,23,24,25]. The increased permeabilization of cell membranes at low temperatures is referred to as the “cold shock phenomenon” and has been ascribed to properties of the cell membrane, implying that the applicability of cold solvents should be evaluated individually for species with different cell wall and membrane properties [24,26,27]. Rabinowitz and colleagues have previously applied a tailored filter-culture protocol to circumvent leakage in *Escherichia coli* [28], but this approach is not compatible with liquid cultures. 

Our lab has specialized in targeted hyphenated MS methods for accurate quantification of the intermediates of central carbon metabolism [3,29,30,31]: a set of evolutionary conserved and interconnected metabolic reactions carrying the majority of metabolic flux for energy conversion and biosynthesis [1]. We have previously applied these methods to quantify and report the central carbon metabolome of eight common eukaryotic and prokaryotic model systems including the Gram-positive bacterium *Bacillus subtilis* [3]. We later extended this with a study exploring metabolite pool dependency on growth rate and cultivation conditions in *Saccharomyces cerevisiae* [32,33]. The Gram-negative and widely used bacterium *E. coli* was, however, not included in the panel in [3] as present fast filtration protocols did not yield the accuracy nor precision required for reporting quantitative metabolite data. Accuracy was evaluated by means of the adenylate energy charge (AEC), a ratio relating the levels of the energy-carrying adenine nucleotides AMP, ADP, and ATP, i.e., the energetic status of a cell [34]. In growth phase, many organisms maintain their AEC within a narrow physiological range of 0.7 to 0.95 [35]. We recorded an average AEC of 0.72 in *B. subtilis*, yet only of 0.43–0.61 when the same protocol was applied to sample and extract *E. coli* [3]. This is far below previous reports [36], and indicates that the sampling or extraction protocol itself is causing ATP turnover to an extent that could possibly mask biologically significant variation. Inconsistent literature reports of intracellular ATP have, in fact, been suggested to be a direct consequence of incomplete quenching and extraction of this highly reactive metabolite [37].

We herein describe the development of a fast filtration-based sampling, quenching, extraction, and sample purification protocol tailored for *E. coli.* We have investigated filter materials and adsorption of extracellular components to the filter, possible membrane permeabilization and metabolite leakage, measured cell geometry and size to avoid size-exclusion, and evaluated the need for and effect of removing extracellular and intracellular components for different types of chromatography. The extensive validation procedure allows us to circumvent common pitfalls of metabolomics sample preparation. It can also be employed for validation of metabolomics protocols for other species of interest. The final fast filtration-based sampling and extraction protocol is relatively simple, relies on commercially available equipment, and can easily be adapted to accommodate different cell densities during time-course experiments. A quick rinsing step tolerated by *E. coli* ensures compatibility with rich medium cultivations by removal of extracellular components which would otherwise compromise the accuracy in reporting of intracellular species, and of salts that are detrimental, especially to ion chromatography. Lastly, we have tailored an efficient extraction solvent that stabilizes a broad range of metabolites, enables sample storage, and is compatible with sample concentration for a variety of downstream hyphenated MS-based metabolic profiling approaches. We demonstrate high precision and low metabolite turnover, and, followingly, minimal perturbation of the inherent metabolic state. This ensures accurate reporting of intracellular metabolite concentrations in *E. coli*. The protocol is also directly transferrable to non-targeted metabolomics, in which high accuracy is a prerequisite for comparing samples, but is harder to monitor. Finally, we apply the protocol to demonstrate precise reporting of the *E. coli* central carbon metabolome under three different commonly applied culture conditions.

## 2. Materials and Methods

### 2.1. Bacterial Strain and Cultivation Conditions

*E. coli* K12 MG1655 (ATCC^®^700926™, ATCC, Manassas, VA, USA) was used for all experiments and cultivated in either rich or mineral media, in either shake flasks or bioreactors.

Overnight (ON) cultures were prepared by inoculating 100 µL of *E. coli* glycerol stock in 100 mL of either rich or mineral culture media in 500 mL baffled shake flasks (2543-00500, Bellco Glass, Vineland, NJ, USA) and incubated with continuous stirring (200 rpm, 37 °C, 16 ± 1 h).

Shake flask cultures were inoculated from ON cultures to an OD_600_ of 0.1 in 500 mL baffled shake flasks, and incubated with continuous stirring (200 rpm, 37 °C) in either rich or mineral media. The rich medium was prepared in MQ-water (18.2 MΩ·cm), with 10 g/L tryptone (T9410, Sigma-Aldrich, Saint-Louis, MO, USA), 5 g/L NaCl (27810.295, VWR, Radnor, PA, USA) and 5 g/L yeast extract (92144, Sigma-Aldrich). The shake flask mineral medium was prepared in MQ-water with 100 mL/L 10× mineral medium salt solution, 0.2465 g/L MgSO_4_·7H_2_O (M5921, Sigma-Aldrich), 4 g/L glucose (101176K, VWR), 2 mL/L trace element solution, and 2 mL/L cobalt-solution (50 mg/L, C8661, Sigma-Aldrich). The 10× mineral medium salt solution was prepared by dissolving 112 g/L Na_2_HPO_4_·7H_2_O (S9390, Sigma-Aldrich), 30 g/L KH_2_PO_4_ (P5655, Sigma-Aldrich), 5 g/L NaCl, and 10 g/L NH_4_Cl (A9434, Sigma-Aldrich), and adjusting the pH to 7.2 with 1 M NaOH (28244.295, VWR). The trace element solution was prepared by dissolving 10 g/L FeSO_4_·7H_2_O (F8633, Sigma-Aldrich), 2.25 g/L ZnSO_4_·7H_2_O (Z0251, Sigma-Aldrich), 2 g/L CaCl_2_·2H2O (223506, Sigma-Aldrich), 1 g/L CuSO_4_·5H_2_O (197722500, ThermoFisher Scientific, Waltham, MA, USA), 0.38 g/L MnCl_2_·4H_2_O (M5005, Sigma-Aldrich), 0.14 g/L H_2_BO_3_ (B6768, Sigma-Aldrich), and 0.1 g/L (NH_4_)_6_Mo_7_O_24_·4H_2_O (1.01182, Sigma-Aldrich) in 5 M HCl (20248.290, VWR).

Bioreactor cultivations were performed in stirred glass autoclavable benchtop bioreactors (1 L, Applikon Biotechnology, Delft, The Netherlands) controlled by my-Control units (Z310210011, Applikon Biotechnology), and equipped with AppliSens Low Drift DO_2_-sensors (Z010023525, Applikon Biotechnology) and AppliSens pH^+^ sensors (Z001023551, Applikon Biotechnology). The bioreactor mineral medium was prepared in the reactors with 5 g/L NH_4_Cl, 2 g/L K_2_HPO_4_ (P8281, Sigma-Aldrich), and 0.5 g/L NaCl in MQ-water and autoclaved. After autoclaving the reactors, the medium was supplemented with 10 g/L glucose, 0.74 g/L MgSO_4_·7H_2_O, 2 mL/L trace element solution, and 2 mL/L cobalt-solution, the last two prepared as described for shake flask mineral medium. The bioreactors were inoculated with 1% (*v/v*%) inoculum from an ON culture in mineral medium to a total volume of 0.9 L. The temperature was set to 37 °C, and the pH was kept constant at 7 by automatic pumping of 4M NaOH. The bioreactors were aerated by sparging air at a flow rate of 600 mL/min. Stirring was adjusted between 200 and 600 rpm throughout the cultivation to keep the level of dissolved oxygen above 30%.

The OD_600_ was measured at regular intervals during cultivation. Growth rates were calculated by exponential regression from three to four data points recorded during exponential growth.

### 2.2. Sampling by Conventional Cold Methanol Quenching

*E. coli* was sampled and extracted by conventional “cold methanol quenching”, as described in [21], with and without washing. Culture broth (5 mL, OD_600_ = 1) was rapidly transferred to a centrifuge tube containing 4 × the volume of cold aqueous methanol (−30 °C, 60%, *v/v*%), immediately vortexed, left shortly at −30 °C and pelleted (4500× *g*, −9 °C, 5 min). The pellet was resuspended in NaCl (0.85%, *w/v*%) for microscopy or washed by resuspension in cold aqueous methanol (−30 °C, 60%, *v/v*%) and pelleted again (4500× *g*, −9 °C, 5 min) before resuspension in NaCl (0.85% *w/v*%). Samples were next prepared for Syto9/PI-staining and assessment of membrane integrity by fluorescence microscopy as described below.

### 2.3. Fluorescence Microscopy for the Assessment of Cell Morphology, Size, and Membrane Integrity

Cell morphology, size, and membrane integrity were evaluated using a Zeiss Axio Imager Z2 microscope equipped with Plan-Apochromat objectives and an Axiocam MR R3 (ZEISS, Jena, Germany) for image capture. Membrane integrity was assessed by means of the LIVE/DEAD BacLight Bacterial Viability Kit (L7012, ThermoFisher Scientific). *E. coli* culture broth (OD_600_ = 1) was pelleted (10,000× *g*, room temperature, 5 min), washed twice in NaCl (0.85% *w/v*%), and stained according to the recommendations of the supplier: 3 µL of a dye mixture containing equal volumes of PI and Syto9 per mL of cell suspension, succeeded by incubation in the dark for 15 min. A total of 5 µL of cell suspension was mounted on PBS-agarose pads (1% *w/v*%. BR0014G, ThermoFisher Scientific; 0710, VWR) for microscopy [38,39]. Images were analyzed with ZEN 2.3 Pro software (ZEISS). Cell width (w) and length (l) were measured from phase contrast images, using calibrated scale bars obtained from the imaging software. Cell volumes were calculated assuming a cylindrical cell-shape capped with two half-spheres, applying the formula V = π × w^2^ × (l − w/3)/4 [40].

### 2.4. Sampling by Fast Filtration and Rinsing of Extracellular Components

Fast filtration of *E. coli* broth was performed applying the technical setup described in [41]: A filtration manifold (X516-1038, VWR) equipped with a magnetic filter funnel (516-7597, VWR) controlled by a vacuum pressure controlling unit (CVC3000 and VSK3000, Vacuubrand, Wertheim, Germany) connected to a vacuum pump (ME 4R NT, Vacuubrand). The filtration manifold was hosed to a glass bottle for collection of flow-through. The glass bottle was further hosed to the vacuum pressure controlling unit with an air filter with a pore size of 0.2 µm (516-7600, VWR) mounted in between to prevent liquid from entering the unit. For plate colony count experiments, the filtration manifold was replaced by a plugged Büchner flask connected directly to the air filter and vacuum controlling unit through the hose barb of the flask. The neck of the magnetic filter funnel was left protruding through the Büchner flask plug and into a 50 mL centrifuge tube placed inside the flask for collection of flow-through. In the latter setup, the vacuum was controlled by clamping the hose.

Culture broth from *E. coli* (OD_600_ = 1) was sampled under a controlled vacuum of 800 mbar below the ambient pressure on hydrated hydrophilic membrane disc filters with a diameter of 47 mm. The filter material, pore size, and supplier of all tested filters are listed in Table 1. All filters were activated with water according to the manufacturer’s instructions. Filtered *E. coli* culture broth was either left unrinsed or rinsed of salt and extracellular components on the filter before it went dry. Then, 10 mL of either cold (~4 °C) or warm (37 °C) MQ-water or shake flask mineral medium without glucose (referred to as ionic rinse) was applied for rinsing. Media background samples were cleared of *E. coli* by centrifugation (4500× *g*, 4 °C, 5 min) and fast filtered and rinsed as described for *E. coli* culture broth samples.

### 2.5. Plate Colony Counts for Assessment of Bacteria in Filtered Flow-Through

Plate colony counts were performed by the spread-plate method. A total of 100 µL of diluted culture or 250 µL of concentrated filtrate was plated on Petri dishes (⌀ 9 cm) containing LB agar (10 g/L tryptone, 5 g/L yeast extract, 5 g/L NaCl, and 15 g/L bacteriological agar (LP0011, ThermoFisher Scientific). The plates were incubated at 37 °C for 16 ± 1 h before colonies were counted.

### 2.6. Rapid Quenching of Metabolic Reactions Post Filtration

A set of forceps was used to carefully transfer the membrane disc filter with filtered *E. coli* kept under a thin intact liquid layer of residual rinsing solution (or broth for unrinsed samples) and submerge it into a cold quench solution. A 10 mL quench solution in a 50 mL centrifuge tube, completely covering the membrane disk filters, was used consistently throughout the rounds of optimization. The filters were quenched in either of the following solvent mixtures; ACN (83640.320, VWR):water (1:1, *v/v*%) kept at ~4 °C on wet ice, or ACN:water:methanol (1.06035.2500, Sigma-Aldrich, 3:5:2, *v/v*%) kept at −20 °C in a low temperature circulating bath (GR150, Grant Instruments, Royston, UK) filled with ethanol. Quenched samples were either left at −20 °C for immediate passive cold extraction, or snap frozen in LN_2_ and stored at −80 °C awaiting extraction.

### 2.7. Metabolite Extraction, Concentration, and Purification

Metabolites were extracted from quenched *E. coli* by either of the following protocols: three repeated freeze-thaw cycles, as described in [3], or 30 min of passive cold (−20 °C) extraction in a low temperature circulating bath filled with ethanol, interrupted by occasional vortexing. Filters were removed, and extracts were cleared of cell debris by centrifugation (4500× *g*, −9 °C, 10 min). Metabolites were concentrated by lyophilization (−105 °C, 0.05 mbar, SP VirTis BenchTop Pro, SP Scientific, Warminster, PA, USA) and reconstituted in 0.5 mL cold MQ-water. The extract matrix was cleared of residual cell debris by centrifugation (4500× *g*, 4 °C, 5 min) and of molecules above either 3 or 10 kDa by centrifugation in 3 (516-0228, VWR) or 10 kDa (516-0230P, VWR) size-cutoff spin filters, respectively (20,800× *g*, 4 °C, 30 or 10 min).

For assessing the effect of lipid-removal, re-constituted extracts were added 0.5 g of hydrophobic C_18_-material (WAT036915, Waters, Millford, MA, USA) and left at 4 °C for 5 min before cell debris and sorbent were cleared by centrifugation in 3 kDa size-cutoff spin filters, as described above.

### 2.8. Flow Cytometric Measurements of Cell Density

*E. coli* culture broth (OD_600_ = 1) was pelleted (10,000× *g*, room temperature, 5 min) and diluted 1:1000 in NaCl (0.85%, *w/v*%). Diluted cell suspensions were stained by applying the LIVE/DEAD BacLight Bacterial Viability Kit according to the recommendations of the supplier, as described for fluorescence microscopy. Stained samples were analyzed on an Attune™ NxT Flow Cytometer (ThermoFisher Scientific) at a flow rate of 25 µL/min. Syto9 was detected with a 488 nm laser and a 530/30 BP filter, while PI was detected with a 561 nm laser and a 620/15 BP filter.

### 2.9. Mass Spectrometric Quantification of Intracellular Amino Acids, Organic Acids, and Phosphorylated Metabolites

Intracellular amino acids, organic acids, and phosphorylated metabolites in cleared cell extracts were prepared for and absolutely quantified by two targeted liquid chromatography (LC)- and capillary ion chromatography (capIC)- tandem mass spectrometry (MS/MS) methods, as described in [3] and [29], with the modifications described in [30]. Absolute quantification was performed by isotope dilution and interpolation from calibration curves prepared from analytical grade standards (Sigma-Aldrich and Santa-Cruz Biotechnology, Dallas, TX, USA) calculated by least-squares regression with 1/x weighting. The AEC was calculated from absolute intracellular concentrations as described in [34].

### 2.10. Mass Spectrometric Quantification of Intracellular Cyclic- and Pyridine Nucleotides

The pyridine nucleotides NAD^+^, NADH, NADP^+^, NADPH, and FAD were sampled, extracted, and quantified by LC-MS/MS from 5 mL of *E. coli* culture broth pelleted at OD_600_ = 1 as described in [31]. Absolute quantification was performed by isotope dilution and interpolation from calibration curves prepared from serial dilution of corresponding analytical grade standards in relevant matrix backgrounds, serving to correct for possible ion suppression. The cyclic nucleotides cAMP and cGMP were sampled, extracted, and analyzed as the pyridine nucleotides. Quantification was performed from the following precursor-product ion transitions: cAMP: m/z ^12^C: 330.114 > 136.065, m/z ^13^C: 340.113 > 141.058, CV: 42 V, CE: 30 eV, and cGMP: m/z ^12^C: 346.114 > 152.005, m/z ^13^C: 356.113 > 157.055, CV: 28V, CE: 22 eV. Absolute quantification of the cyclic nucleotides was performed by isotope dilution and interpolation from a calibration curve prepared from serial dilutions of analytical grade standards of cAMP (A9501, Sigma-Aldrich) and cGMP (G6129, Sigma-Aldrich) calculated by least-squares regression in a pooled-matrix background, as described in [31].

### 2.11. Dry-Weight Measurements

A total of 5 mL of *E. coli* broth (OD_600_ ~ 1) from shake flasks with rich or mineral medium, or from bioreactor with mineral medium were fast filtered through Durapore^®^ filters with a pore size of 0.45 µm and rinsed with 10 mL MQ-water (37 °C), applying the setup described above. Filters were transferred to pre-weighed aluminum pans (611-1376, VWR) and dried in a heating cabinet (110 °C) until constant weight. OD_600_ was plotted against cell dry weight (CDW) per L sampled broth for all three media/cultivation systems, and CDW per L culture broth for individual samples was calculated by interpolation.

### 2.12. Normalization of Metabolite Extract Concentrations and Calculation of Intracellular Metabolite Concentrations

Measured metabolite extract concentrations were corrected for dilution and concentration performed during sample purification and normalized to either experimental cell density (cells/L) as recorded by flow cytometry, or to CDW (g/L) interpolated from the corresponding OD_600_ vs. CDW (g/L) curve. Intracellular metabolite concentrations were calculated by applying experimental cell volumes from the corresponding cultivation system/medium.

## 3. Results and Discussion

### 3.1. Sampling by Conventional “Cold Methanol Quenching” Permeabilizes the E. coli Cell Membrane

Rapid quenching of metabolic activity is of high importance in metabolomics, but the quenching procedure itself should not in other ways perturb the system. To assess the permeability of the *E. coli* cell membrane in cold aqueous methanol and, hence, the applicability of conventional “cold methanol quenching” for this organism specifically, we stained methanol-quenched *E. coli* cultures with the red-fluorescent propidium iodide (PI) and a counterstain. PI is not permeant to intact cells but can enter through compromised membranes and intercalate between the bases of DNA, while the green-fluorescent counterstain Syto9 can enter through intact membranes [42]. The proportions of PI-stained cells in an unquenched culture, a cold methanol-quenched culture, and a cold methanol-quenched culture subjected to a subsequent cold methanol wash were estimated by fluorescence microscopy imaging. The resulting staining ratios confirm that quenching in cold aqueous methanol compromises the *E. coli* cell membrane and that the fraction of compromised cells scales with the residence time in the cold quenching solvent (Figure 1). These data align with previous reports of metabolite leakage from *E. coli* into cold aqueous methanol accompanied by AECs far below the physiological range [2,13], highlighting the need for alternative sampling and quenching protocols for this organism.

Fast filtration is an alternative sampling strategy that offers rapid concentration of biomass prior to quenching. In the following sections, we explore this approach specifically for *E. coli* and perform comprehensive testing to circumvent common pitfalls.

### 3.2. Filter Material and Pore Size Are Critical Parameters in Fast Filtration-Based Sampling Protocols

We and others have previously demonstrated the applicability of controlled fast filtration through hydrophilic polyethersulfone (PES) disk filters for targeted metabolic profiling of several microorganisms [3,43]. When re-evaluating PES filters for an *E. coli* fast filtration-based sampling protocol, we, however, found that this filter material does release components that will dominate high-resolution mass spectra when submerged in organic solvents. More specifically, the Q-TOF MS^E^ total ion chromatograms of “mock metabolite extracts” from PES filters extracted in acetonitrile (ACN):water were dominated by repetitive peaks separated by 44 Da, i.e., the typical polyethylene glycol (PEG) mass distribution (Appendix A). If co-eluting with analytes, such matrix components can cause ion suppression and compromise the quantitative accuracy of MS analyses [44]. We thus concluded that PES filters are incompatible with organic solvent extractions and excluded this filter material from further testing.

The native hydrophobicity of filter materials such as polytetrafluoroethylene (PTFE, Fluoropore) restricts their applicability for filtering aqueous culture media. We therefore focused on hydrophilic filter materials and tested three different types: Polyvinylidene fluoride (PVDF, Durapore), hydrophilic coated PTFE (Omnipore), and nylon. The filters were placed in a magnetic filter funnel on a filtration manifold controlled by a vacuum control unit and a vacuum pump. The pump was operated at 800 mbar below the ambient pressure to ensure a controlled, steady flow of liquid. Initially, a volume of *E. coli* culture broth equivalent to five optical density (OD_600_)-units was filtered and rinsed with 10 mL of Milli-Q (MQ)-water on filters of the three materials and of all available pore sizes below the average diameter of *E. coli*: 1 µm [45]. Total filtration times were recorded, revealing that filtration through PVDF filters was the fastest for all pore sizes (Figure 2a). The performance of specifically the two largest PVDF pore sizes was promising considering previously published fast filtration studies for *E. coli*, in which total filtration times were 30–50% longer [13,46].

The purpose of fast filtration for metabolomics is to efficiently separate all biomass from the surrounding media, i.e., ensure a quick flow of liquid without selecting for subpopulations based on size or shape. To evaluate the applicability of available filter pore sizes for this purpose, we measured the average diameter of *E. coli* cultivated in several common laboratory setups by light microscopy. The resulting size distribution revealed that varying the cultivation method and media does indeed influence the size of *E. coli* (Figure 2b). All tested filter pore sizes (0.22, 0.45, and 0.65 µm) were, however, smaller than the lowest measured diameter of *E. coli* and should retain the complete population during fast filtration, given that the applied vacuum does not affect the shape of the cell wall or membrane.

To test for this, we repeated the experimental setup from Figure 2a with PVDF filters only, this time collecting and plating the filtrate to quantify any potential loss of biomass. Plate colony counts for the unfiltered culture broth were at least 500 times higher than for the filtrate from all PVDF filter pore sizes (Figure 2c, note the different axes scales). This demonstrates that although a small fraction of the *E. coli* population is lost during filtration under controlled vacuum and the size of this fraction correlates with pore size, the loss is less than 0.2% even for the largest pore size and, consequently, neglectable.

As we measured the lowest filtration time (Figure 2a) combined with low bacterial loss (Figure 2c) on PVDF filters with a pore size of 0.45 µm, we decided to move forward with these filters for further testing. Additionally, low protein adsorption has been measured for this hydrophilic polymer [47], and this is favorable for minimizing matrix effects in downstream analyses. Lastly, we verified the absence of contaminating substances leaking from PVDF filters by high-resolution MS (Appendix A).

We detected all expected central carbon metabolites applying our published extraction protocol [3] to biomass filtered on 0.45 µm PVDF filters, indicating that sampling 5 OD_600_ units of *E. coli* is sufficient for MS-based metabolic profiling of central carbon metabolism (data not shown). Doubling the amount of filtered biomass tripled the total filtration time (Figure 2d) and significantly lowered the AEC (Figure 2e), indicating that the maximal amount of biomass applicable for fast filtration is less than 10 OD_600_ units. To allow for time-course experiments and fast filtration at high ODs, we suggest adapting the sample volume throughout the course of an experiment to compensate for changing biomass and to ensure that the filtration time is similar at every sampling point. A changing sampling volume can be compensated for by the simultaneous application of medium to the filter, keeping the total filtration volume constant throughout the course of an experiment. It must be noted that the accuracy and precision in pipetting small volumes will limit the applicability of this approach at very high ODs.

### 3.3. Filtered E. coli Tolerates a Quick Warm Water Rinse Serving to Remove Extracellular Components and Ensure Analytical Performance

Fast filtration efficiently separates biomass from the bulk culture medium, but residual extracellular components will remain on cell surfaces and in the thin liquid layer left on the filter. This can compromise the quantitative accuracy in measurements of corresponding intracellular metabolites and challenge analytical accuracy through interference with downstream chromatography and MS. Though the inclusion of a rinsing step can circumvent this, the complete removal of extracellular components is a metabolic perturbation. Thus, the necessity of doing so should be thoroughly evaluated [37].

To evaluate the need for and effect of rinsing *E. coli* for downstream targeted LC- and IC-MS/MS-based metabolic profiling, we first filtered and rinsed a culture grown in mineral medium with 10 mL of warm (37 °C) mineral medium without glucose (hereafter referred to as ionic rinse), or with warm (37 °C) or cold (4 °C) MQ-water. Comparing total filtration times revealed that cold rinsing was significantly slower than warm rinsing (Figure 3a). Also, LC-MS/MS-based quantification of the proteogenic amino acid pool indicated significantly lower metabolite recovery in cold-rinsed cells, a possible consequence of “cold shock” and membrane permeabilization (Figure 3b). As capIC-MS/MS-based quantification of the adenine nucleotides did not indicate that cold rinsing served to stabilize these high-energy metabolites during sampling (Figure 3c), we did not pursue this strategy any further for *E. coli.*

Rinsing with a solution ionic to the culture medium has been suggested to circumvent leakage for several species of bacteria [13], and initial measurement of amino acids indicated that this rinsing strategy was superior to the others with respect to metabolite recovery (Figure 3b). However, acknowledging that salts can bias downstream MS analyses both through altered derivatization efficiency, electrospray ionization (ESI) adduct formation, and altered chromatographic performance [44], prompted us to carefully inspect the underlying MS raw data. To obtain the highest possible accuracy and precision for metabolic profiling, we consistently apply isotope dilution, i.e., correct each analyte by the corresponding ^13^C, (^15^N)-labeled internal standard (ISTD) spiked into the extract [48]. When reviewing the amino acid data, we found a significant reduction in the MS signal of both analytes and the corresponding ISTDs in extracts of ionically rinsed cells (Figure 3d, left panel). This pattern indicates severe ion suppression, but as the suppression affects both analytes and ISTDs, this phenomenon should be fully corrected by isotope dilution. If, however, the ion suppression occurs to an extent that forces the ISTD signal into the noise range, isotope dilution will no longer improve accuracy but instead confer uncertainty, as the recorded analyte signal is corrected by an uncertain ISTD signal. We found several indications of this occurring: (1) The reduction in ISTD MS signal did not scale with the reduction in the corresponding analyte MS signal (Figure 3d, left panel), and (2) The signal-to-noise ratios (S/N) in extracts of cells subjected to an ionic rinse were significantly lower than in extracts of cells subjected to a water rinse (Figure 3d, right panel). Taken together, this implies that the analyte MS signals are corrected by low and uncertain ISTD signals, which renders the calculated intracellular amino acid concentrations uncertain and artificially high (Figure 3b).

IC separates ion species based on the principle of ion exchange and is particularly sensitive to the presence of salts [29]. Comparing chromatograms from capIC-MS/MS-based analysis of *E. coli* subjected to a warm water and a warm ionic rinse revealed that the presence of salts decreased the retention time, chromatographic resolution, and MS signal of phosphorylated analytes eluting throughout the run (Appendix A). As exemplified by the quantification of the adenine nucleotides, concentrations were seemingly higher in cells subjected to an ionic rinse (Figure 3e). However, the MS signals applied to calculate concentrations with the ionic rinse were lower for both analytes and ISTDs (Figure 3f, right and left panel, respectively), indicating ion suppression. The reduction in analyte and ISTD signals were of the same magnitude both for ADP and ATP, indicating that isotope dilution might be an effective strategy to correct these high-abundant metabolites. However, salts originating from the ionic rinse suppressed the signal of several other phosphorylated metabolites, amongst them AMP, below the limit of detection (LOD, Figure 3f). This rendered capIC-MS/MS-based quantification of the complete phosphometabolome impossible at salt concentrations resulting from ionic rinsing.

Throughout Figure 3, we have demonstrated that the presence of extracellular salts can cause ion suppression to an extent that severely compromises the quantitative accuracy of both LC- and IC-MS/MS-based metabolic profiling, even when applying sophisticated correction strategies. We measured high MS signals in extracts of *E. coli* subjected to a quick rinse with warm water (Figure 3d or Figure 3f), but this rinsing strategy cannot be justified if found to compromise the metabolic state of the organism. To evaluate this, or more specifically, to evaluate possible membrane permeabilization and leakage resulting from rinsing, we stained *E. coli* subjected to a quick warm water rinse with the membrane impermeant dye PI and estimated the stained fraction by fluorescence microscopy imaging. The staining pattern of rinsed filtered cells was similar to that of unrinsed filtered cells (Figure 4), confirming that the *E. coli* cell membrane tolerates a quick, warm water rinse under controlled vacuum.

### 3.4. Combined Quenching and Extraction in Cold Acetonitrile:Water:Methanol Stabilizes High Turnover Metabolites to Allow for High-Precision Metabolic Profiling of E. coli

Even with an optimized filtration-based protocol for sampling, we measured an average AEC far below the reported physiological range of *E. coli* [36] in cold ACN:water samples extracted by repeated freeze-thaw cycles as described in [3,29] (Figure 3c). The low AEC indicates persisting enzyme-catalyzed and/or chemical degradation, which can result from inadequate quenching and/or failure to stabilize metabolites during extraction and sample purification. This hypothesis was substantiated by quantification of 32 intermediates of central carbon metabolism in *E. coli* extracts, revealing a large spread between replicate samples, as exemplified for selected metabolites from different pathways in Figure 5a. The spread and low AEC could not be explained by highly concentrated extracts overloading the column or the presence of interfering lipid species, as neither dilution of extracts nor lipid removal improved the outcome (Figure 5b,c). The deviation between replicates was, however, higher for phosphorylated metabolites than for other pathway intermediates, e.g., the tricarboxylic acid (TCA) cycle intermediate succinate (Suc, Figure 5a). This suggests that sporadic loss of phosphate groups might be the cause of unprecise phosphometabolite measurements. We thus initiated an optimization process to stabilize these high-turnover metabolites in *E. coli* during extraction and subsequent extract processing steps.

Metabolite extract purification procedures such as lipid and protein removal can reduce the overall extent of matrix effects, but may also influence recovery [49,50]. For phosphometabolome profiling, we tested removing lipids by a solid phase extraction (SPE) procedure involving a membrane-based size-exclusion centrifugation step to remove the sorbent. As we measured a very low AEC in *E. coli* extracts subjected to this procedure (Figure 5c, right), we hypothesized that the inclusion of the additional sample purification step could have accelerated ATP turnover. We have previously purified metabolite extracts of proteins by size-exclusion centrifugation with a cutoff of 3 kDa [3,29,32], but did note that this procedure was more time-consuming for *E. coli* than for other microorganisms and human cell lines. To rule out whether slow size-exclusion centrifugation was accelerating metabolite turnover in *E. coli,* we compared the recovery and replicate spread of phosphometabolites in extracts subjected to this procedure with either a low (3 kDa, according to [3]) or higher (10 kDa) size cutoff. Increasing the size cutoff reduced the total centrifugation time (data not shown) and improved both the phosphometabolite recovery (selected metabolites in Figure 5d) and the AEC (Figure 5e), without increasing the capIC or LC column pressure nor reducing overall chromatographic performance. Thus, turnover was lower while extracts remained compatible with downstream hyphenated MS analyses. Shifting the size cutoff did, however, not eliminate the replicate spread, indicating that low precision could not be ascribed solely to extensive sample purification (Figure 5d). This prompted further investigation of alternative quenching and extraction protocols.

The literature is ambiguous with regards to quenching and extraction solvents for metabolomics, and a variety of solvent systems and procedures have been described for the extraction of yeast, bacteria, and mammalian cells (reviewed in [18]). Extraction in cold ACN:water (1:1, *v/v*%) has been found superior to conventional boiling ethanol and cold methanol extraction for both yeast and mammalian cells [14,51], and recovered all expected central carbon metabolites from *E. coli* quenched in liquid nitrogen (LN_2_) and subjected to repeated freeze-thaw cycles (data not shown). Despite this, the cold ACN:water solvent system seemingly failed to fully stabilize central carbon metabolites in *E. coli* during the course of a standard freeze-thaw extraction regime (Figure 5a–d). This indicates that optimal quenching and extraction conditions are species-specific, as also discussed by others [14,15]. Cold (−48 °C) methanol and multiple freeze-thaw cycles have previously been recommended for metabolite extraction from *E. coli* [27], and we hypothesized that the low temperature of this system might serve to stabilize phosphometabolites in whole-cell extracts. The favorable low melting point of methanol does, however, also render it incompatible with solvent removal by lyophilization. Solvent removal is beneficial for metabolomics for several reasons: it offers concentration of dilute extracts and improved stability upon storage [50], it ensures compatibility with all downstream hyphenated MS analyses, and it is tolerated by most central carbon metabolites, including ATP [37]. The alternative strategy, methanol evaporation under vacuum, is inefficient for removing the large volumes required to cover and quench a 47 mm membrane disk filter. We thus started investigating cold extraction solvent mixtures compatible with lyophilization.

Rabinowitz and Kimball have previously examined several solvent systems for extraction of water-soluble metabolites from *E. coli*, only to conclude that amino acids and several central carbon metabolites are adequately extracted by a broad spectrum of solvents systems [16]. They did, however, note that ACN:water:methanol (2:2:1, *v/v*%) gave superior triphosphate yields, indicating that this solvent system efficiently stabilizes such high-turnover metabolites, as confirmed in [52]. Inspired by this, we found that combining ACN:water and methanol at a ratio of 3:5:2 (*v/v*%) allowed us to exploit the benefits of all three solvents. While the combined solvent system remained in a liquid state at −20 °C enabling cold extraction in a standard stirred ethanol bath, it could be frozen in LN_2_ for temporary storage, and the small methanol fraction quickly evaporated under vacuum in a lyophilizer without affecting sublimation from the ACN:water fraction. Fluorescence microscopy imaging revealed that 30 min of passive cold (−20 °C) extraction in ACN:water:methanol (3:5:2, *v/v*%) permeabilized the cell membranes of ≥95% of the total *E. coli* population (Figure 5f). Hence, this simple extraction regimen could serve to bypass laborious extraction protocols involving repeated freeze-thaw cycles, as previously applied by us and others [3,27]. This was confirmed from the recovery of nucleotides, and phosphorylated intermediates of glycolysis and the pentose phosphate pathway (PPP) extracted in this solvent system, which was even higher in samples subjected to passive cold extraction than to three consecutive freeze-thaw cycles (Figure 5g). Finally, comparing the replicate spread and AEC of *E. coli* extracted in cold (−20 °C) ACN:water:methanol (3:5:2, *v/v*%) to that of *E. coli* extracted in cold (4 °C) ACN:water established that the former optimized solvent system efficiently increased precision and stabilized high-turnover metabolites (Figure 5h,i). In other words, it minimized replicate spread to allow for high-precision metabolic profiling of *E. coli* with an AEC well within the reported physiological range.

### 3.5. Optimized Sampling, Extraction, and Sample Purification Enable Precise Quantification of the E. coli Central Carbon Metabolome under Different Growth Conditions

To demonstrate the applicability of the optimized sampling, extraction, and sample purification protocol developed as described throughout the previous sections, we profiled and compared the central carbon metabolome of exponentially growing *E. coli* MG1655 cultivated in three commonly applied laboratory setups; shake flasks and stirred benchtop bioreactors, the former both with mineral and rich media. The optimized protocol relies on lyophilization and is, hence, compatible with all solvent systems for downstream hyphenated MS analyses. We achieved broad coverage of central carbon metabolism by concentrating the extracts in two different solvent systems for metabolic profiling by capIC-MS/MS and reverse phase (RP) LC-MS/MS with upfront derivatization (Figure 6a). Though solvent removal is tolerated by most central carbon metabolites, it can induce oxidation of redox-active metabolites [37]. We thus performed sampling and extraction of the important redox co-factors NAD(H) and NADP(H) by a complementary method tailored to accommodate these metabolites [31].

Like *S. cerevisiae* and *B. subtilis* profiled and described in [3], *E. coli* was sampled in mid-exponential phase (OD_600_ = 1), at the growth rates summarized in Figure 6b (corresponding growth curves plotted in Appendix A). Absolute extract concentrations were calculated by isotope dilution and interpolation from calibration curves prepared from serial dilutions of analytical standards. Intracellular concentrations were next calculated by normalization to experimental cell counts obtained from flow cytometry and experimentally determined cell sizes obtained by microscopy image quantification. The latter allowed us to account for culture condition-dependent variation in cell size (Figure 2b and [53]). Reporting absolute intracellular concentrations provides an additional, critical layer of information to ion intensities and relative fold changes. It enables direct and biologically contextualized interpretation of results, as the measurements can be related to biochemical parameters such as the Michaelis–Menten constant (K_M_) of relevant enzymes [54,55]. Further, these quantitative data can be exploited for relevant mathematical modeling, such as Metabolic Control Analysis [56] or thermodynamic calculations. As relating metabolite abundance to CDW or OD is less laborious, these alternative normalization strategies are also commonly applied in metabolomics [54,57,58]. To investigate whether the choice of normalization strategy would strongly influence the interpretation of the resulting data, we also performed normalization to CDW, and concluded that most trends were conserved across both normalization strategies (Appendix A).

In total, 73 central carbon metabolites were absolutely quantified in *E. coli* extracts, including intermediates of glycolysis, the PPP and TCA cycle, amino acids, and deoxy-, cyclic, and redox-active- and standard nucleotides. Such comprehensive datasets are scarce, even in databases dedicated to reporting the metabolome of specific organisms, providing a limited basis for comparing intracellular concentrations across studies. Several studies of *E. coli* have, however, reported that intracellular metabolite concentrations are dependent on available carbon sources [55,59,60]. By comparing the metabolite profiles of *E. coli* cultured in media with different nutrient composition and with and without control of dissolved oxygen and pH, as presented herein, we could explore and add to the understanding of how these important parameters affect central carbon metabolism.

The intracellular metabolite concentrations in *E. coli* grown in rich medium spanned five orders of magnitude, from 10^−2^ mol/L to 10^−6^ mol/L. Concentrations were lower in cells grown in mineral media, spanning from 10^−3^ mol/L to 10^−6^ mol/L (color-coded according to concentration range in Figure 6a and listed in Appendix A). While the high-abundant metabolites (10^−2^ mol/L) in *E. coli* cultured in rich medium were all amino acids, these were present only in sub-millimolar concentrations when *E. coli* was cultured in mineral media, leaving the total difference in amino acid content more than twenty-five-fold (Figure 6c). High intracellular amino acid levels in rich medium cultivations have previously been reported both for *S. cerevisiae*, *B. subtilis,* and *E. coli* [3,59], and can be ascribed to amino acid uptake from the medium. Correspondingly low levels of biosynthetic intermediates have also been reported for rich medium cultivations of *E. coli* [60]. Oppositely, as mineral media do not contain amino acids, the intracellular amino acid pool in these cultivations originates solely from de novo synthesis, which is consistent with a lower intracellular reservoir (Figure 6a,c). To make sure the intracellular amino acid pool size of *E. coli* cultured in rich medium was not biased by adsorption of extracellular amino acids to the sampling filter material, we measured and subtracted the amino acid concentrations in spent media from the concentration measured in cell extracts. Correction for possible adsorption of extracellular species did not change the overall distribution between metabolite classes, verifying the measurements of intracellular concentrations (Appendix A).

The total magnitude of the other metabolite classes and pathway intermediates were more conserved across culture conditions (Figure 6c), though several differences were evident at the single-metabolite level (Figure 6a). This is consistent with a previous report by Bennett et al. [55], concluding that available carbon sources have a substantial impact on the *E. coli* metabolome, but does not reshape the overall metabolome composition due to the low abundance of many of the changing metabolites. The metabolites found to differ the most were downstream of assimilation of the carbon source in question, which is consistent with our reports of higher levels of glycolytic and PPP intermediates in mineral media with glucose compared to the rich medium (Figure 6a and Appendix A). Elevated levels of intracellular sugars have also been reported by others [59] for glucose-fed *E. coli*.

Concluding that the overall metabolic landscape of *E. coli* is quite homeostatic across cultivation conditions in exponential phase, we decided to focus on a few selected metabolites of high importance. Examining the concentrations and ratios of high-energy adenine nucleotides in *E. coli* revealed that both the AEC and the ATP pool was maintained at a higher level in mineral media cultures compared to the rich medium culture (Figure 6d,e), indicating that high glucose availability could allow the cells to maintain a higher energy reserve. Also, though a high AEC was maintained independent of cultivation vessel in mineral media (Figure 6d), quantification of the total adenine nucleotide pool indicated that this required a higher adenine nucleotide reserve when the level of dissolved oxygen was not controlled, as in shake flasks (Figure 6e). To explore the dependency of the redox state of *E. coli* on cultivation conditions, we investigated the ratio of reduced to oxidized NAD and found it to differ significantly between cultivation conditions (Figure 6f). Though the size of the intracellular NAD pool was similar in both mineral media cultivations (Figure 6g), the NADH/NAD^+^ ratio was significantly higher in shake flasks (Figure 6f). This may be a consequence of lower oxygen availability in the shake flasks, and thus a lack of electron acceptor for the electron transport chain, causing accumulation of NADH from glycolysis and the TCA cycle. The NADH/NAD^+^ ratio was, however, maintained at a low level when *E. coli* was cultured in rich medium under the same level of oxygenation (Figure 6g), possibly due to a lower basal level of NAD reduction in rich medium than in the mineral medium.

## 4. Conclusions

We have previously profiled and reported the central carbon metabolome of eight common model organisms [3]. The metabolome of the Gram-negative bacteria *E. coli* was, however, not reported due to insufficient precision and accuracy, indicating that the sampling and extraction procedure for downstream metabolic profiling of this organism requires specific attention. Throughout this study, we have demonstrated how optimizing parameters including filter material and pore size, biomass per sample, the temperature and composition of rinsing, quenching, extraction solvents, and sample purification steps ensures high-precision metabolic profiling of *E. coli* with low metabolite turnover and hence, minimal impact on the inherent metabolic state. The resulting sampling, extraction, and sample purification protocol for metabolic profiling of *E. coli* is summarized in Figure 7, highlighting critical considerations and parameters together with recommended values, materials, and procedures, as discussed throughout this study.

By applying this protocol to *E. coli* cultured under three different cultivation conditions we demonstrated that varying the media composition and control of dissolved oxygen at pH does not reshape the overall molar composition of central carbon metabolism at a low optical density, except for the amino acid pool. Rather, changes can be observed at the single-metabolite level. This also applies to levels of important redox and energy-carrying metabolites.

## Figures and Tables

**Figure 1 metabolites-13-00150-f001:**
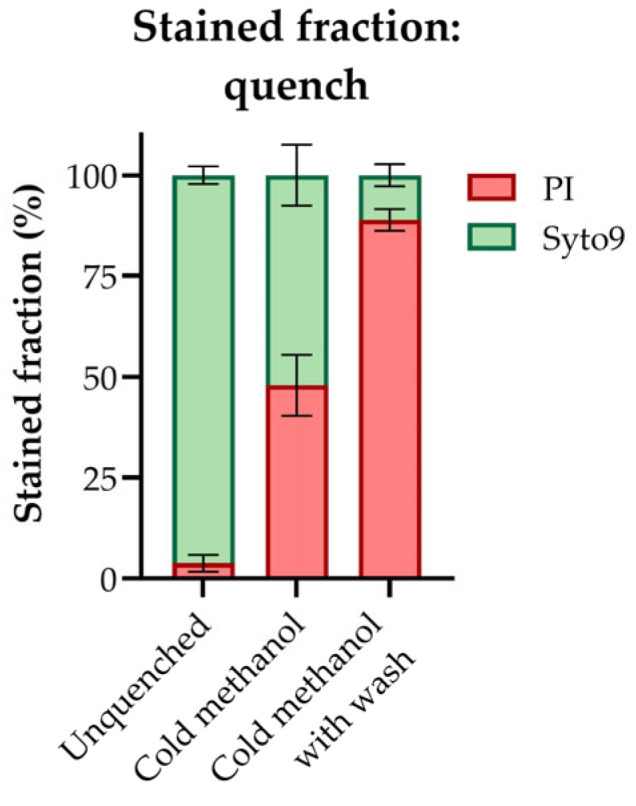
The fraction of propidium iodide (PI) and Syto9 stained cells in an unquenched *E. coli* culture and a cold methanol-quenched (−30 °C) *E. coli* culture with and without subsequent washing in cold methanol (−30 °C), as estimated by fluorescence microscopy imagining. The average ± SD of n = 5 technical replicates is presented. More than 50 cells were counted per technical replicate.

**Figure 2 metabolites-13-00150-f002:**
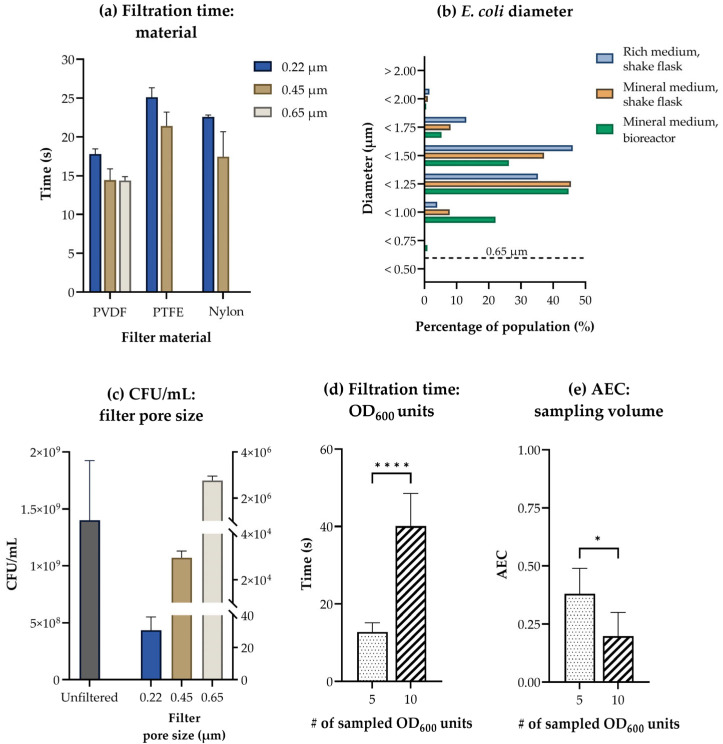
Effects of filter material, pore size, and sampled biomass. (**a**) Filtration times (s) for five OD_600_ units of *E. coli* rinsed with 10 mL warm (37 °C) water under a controlled vacuum on polyvinylidene difluoride (PVDF), hydrophilic polytetrafluoroethylene (PTFE), and nylon filters with a pore size of 0.22 µm, 0.45 µm, or 0.65 µm. The average ± SD of n = 3–4 technical replicates is presented. (**b**) The proportions of *E. coli* from three common laboratory setups belonging to different diameter-ranges (µm), as estimated by light microscopy. Presented for rich medium in shake flask, mineral medium in shake flask, and mineral medium in bioreactor. The largest filter pore size included in this study is indicated by a dashed line. The average of 1–3 biological replicates, each with n ≥ 5 technical replicates is presented. More than 350 cells were measured per biological replicate. (**c**) Colony forming units (CFU)/mL in five OD_600_ units of an unfiltered *E. coli* culture, and in the corresponding *E. coli* filtrate from PVDF filters with a pore size of 0.22 µm, 0.45 µm, and 0.65 µm. The average ± SD of n = 3 technical sampling replicates each plated in 1–3 parallels is presented. (**d**) The filtration time (s) and (**e**) adenylate energy charge (AEC) measured for five and ten OD_600_ units of *E. coli* rinsed with 10 mL warm (37 °C) water under a controlled vacuum on PVDF filters with a pore size of 0.45 µm. The average of n = 4–8 replicate samplings is presented. Significance levels from two-tailed *t*-tests assuming equal variances are marked in (**d**,**e**) *: *p* ≤ 0.05; ****: *p* ≤ 0.0001. #; number.

**Figure 3 metabolites-13-00150-f003:**
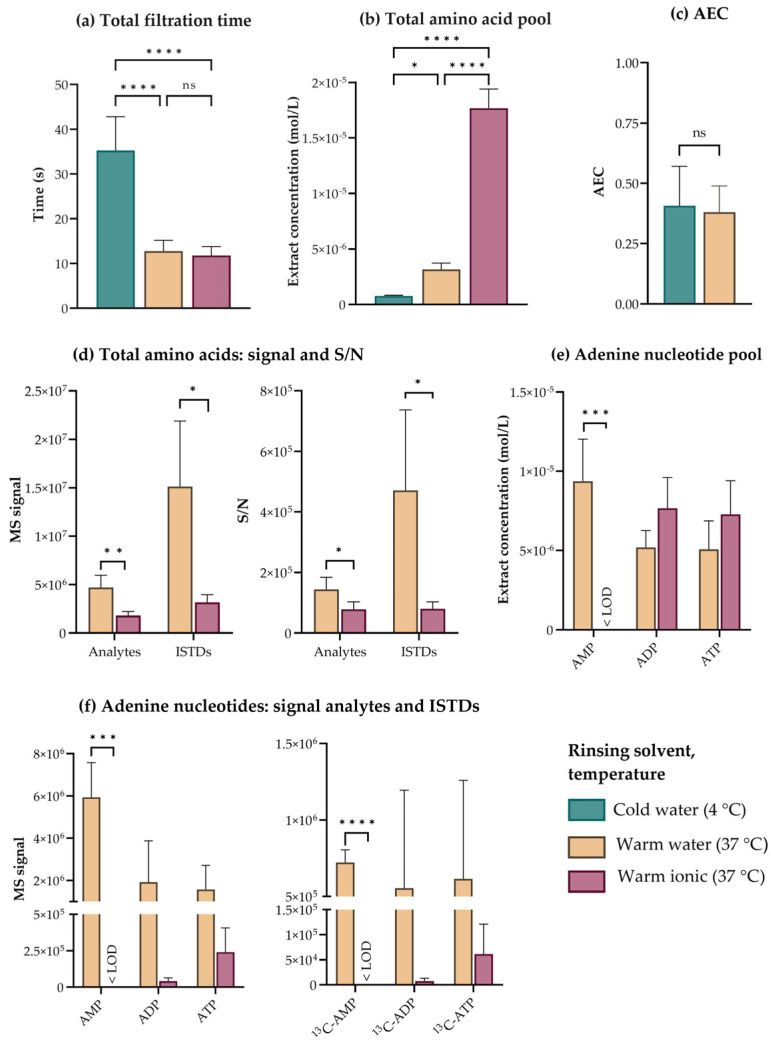
Effects of rinsing solvent and temperature. (**a**) Filtration times (s), (**b**) total amino acid extract concentrations (mol/L), and (**c**) adenylate energy charges (AECs) as measured for five OD_600_ units of *E. coli* rinsed with 10 mL of cold water (4 °), warm water (37 °C), or a warm rinsing solution ionic to the culture medium (37 °C) under a controlled vacuum on polyvinylidene difluoride (PVDF) filters with a pore size of 0.45 µm. (**d**) Total mass spectrometric (MS) signal (left panel) and signal-to-noise ratio (S/N) (right panel) of the total amino acid pool and corresponding ^13^C, (^15^N)-labeled internal standard (ISTDs), (**e**) adenine nucleotide extract concentrations (mol/L), and (**f**) MS signals (left panel) and S/N (right panel) of the adenine nucleotides and corresponding ISTDs, all in five OD_600_ units of *E. coli* rinsed with 10 mL of water warm water (37 °C) or a warm rinsing solution ionic to the culture medium (37 °C) under a controlled vacuum on PVDF filters with a pore size of 0.45 µm. The average ± SD of n = 4–8 technical replicates is presented. Significance levels for multiple comparisons from one-way ANOVA with Tukey’s multiple comparisons tests (**a**,**b**) and for pairwise comparisons are from two-tailed *t*-test assuming equal variances (**c**–**f**), are both marked ns: not significant; *: *p*≤ 0.05; **: *p* ≤ 0.01; ***: *p* ≤ 0.001; ****: *p*≤ 0.0001. LOD; limit of detection.

**Figure 4 metabolites-13-00150-f004:**
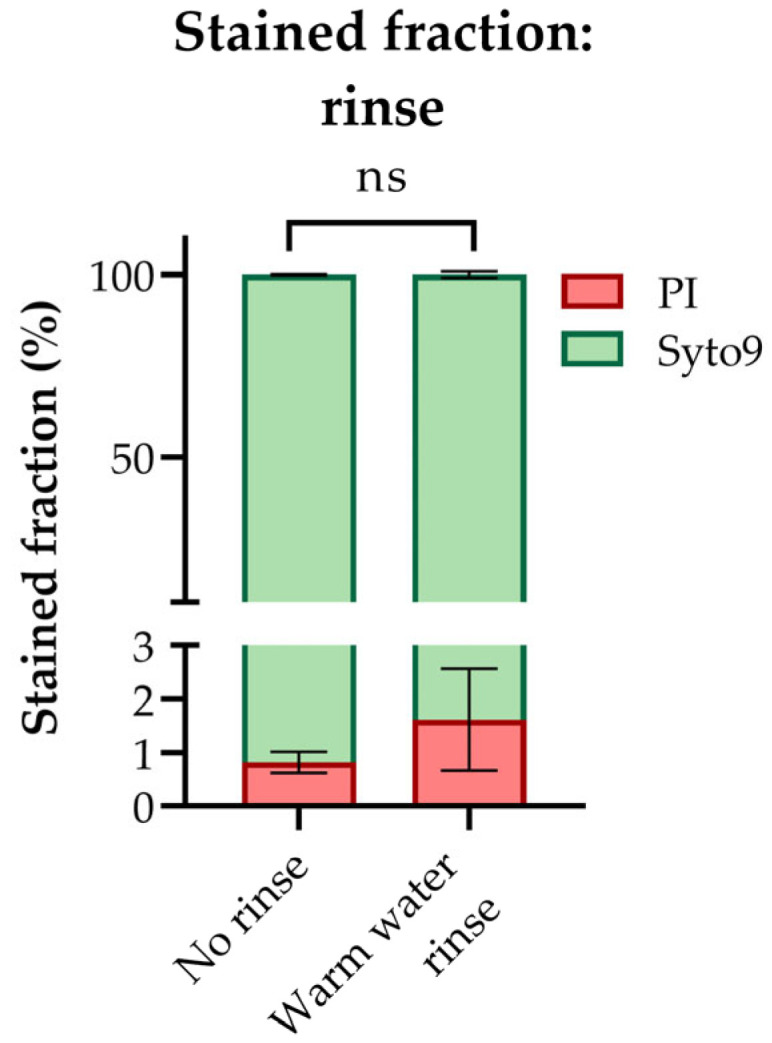
The fraction of propidium iodide (PI) and Syto9 stained cells in an *E. coli* culture sampled by fast filtration under controlled vacuum with and without a quick warm (37 °C) water rinse on the filter, as estimated by fluorescence microscopy imagining. The average ± SD of n = 3 technical replicates is presented. >3000 cells were counted per technical replicate. Significance levels from two-tailed *t*-tests assuming equal variances are marked ns: not significant.

**Figure 5 metabolites-13-00150-f005:**
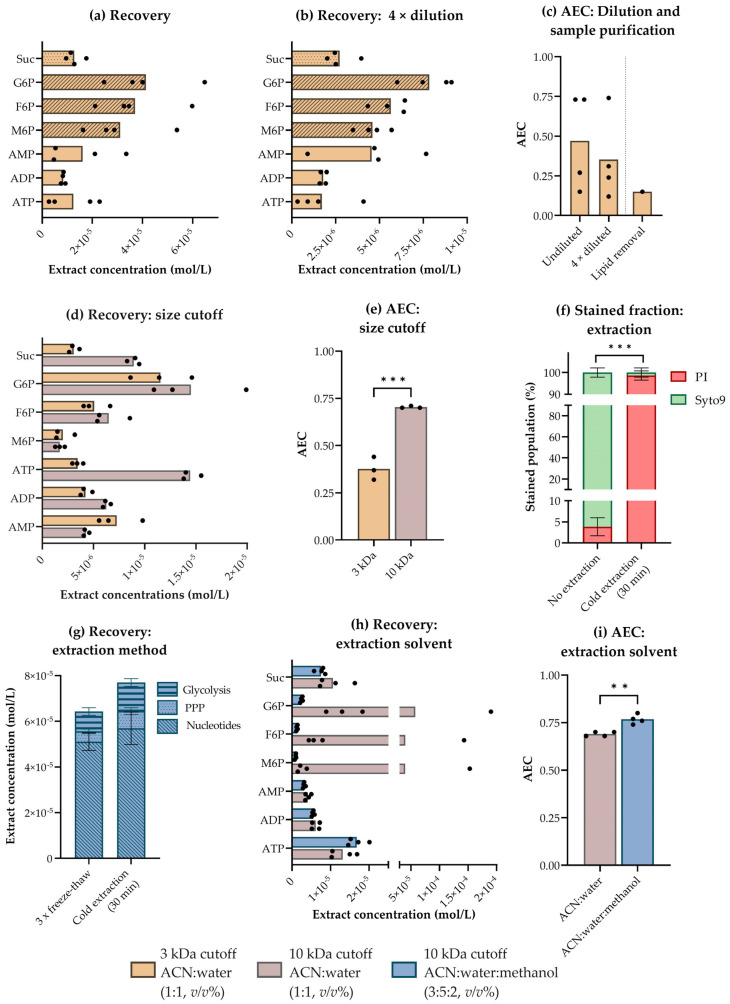
Recovery and stability in *E. coli* metabolite extracts subjected to different extraction and sample purification procedures. Extract concentrations (mol/L) of selected intermediates of central carbon metabolism in (**a**) undiluted and (**b**) four-times diluted acetonitrile (ACN):water (1:1, *v/v*%) *E. coli* extracts cleared by size-exclusion centrifugation with a cutoff of 3 kDa. (**c**) Adenylate energy charges (AECs) in undiluted and four-times diluted *E. coli* extracts (left panel) and in *E. coli* extracts subjected to a solid phase extraction (SPE) lipid removal procedure involving size-exclusion centrifugation (right panel), all extracted in ACN:water (1:1, *v/v*%) and cleared by size-exclusion centrifugation with a cutoff of 3 kDa. (**d**) Extract concentrations (mol/L) of selected intermediates of central carbon metabolism and (**e**) AECs as measured in *E. coli* extracted in ACN:water (1:1, *v/v*%) and cleared by size-exclusion centrifugation with a cutoff of 3 or 10 kDa. For (**a**–**e**), the average and spread of n = 3–4 technical replicates are presented. (**f**) The fraction of propidium iodide (PI) and Syto9 stained cells in a non-extracted *E. coli* culture and an *E. coli* culture subjected to passive cold extraction (−20 °C, 30 min) in ACN:water:methanol (3:5:2, *v/v*%, 30 min), as estimated by fluorescence microscopy imagining. The average ± SD of n = 5 technical replicates is presented. > 200 cells were counted per technical replicate. (**g**) Extract concentrations (mol/L) of nucleotides and intermediates of glycolysis and the pentose phosphate pathway (PPP) in *E. coli* extracted by three repeated freeze-thaw cycles or by passive cold extraction (−20 °C, 30 min), both in ACN:water:methanol (3:5:2, *v/v*%). The average ± SD of n = 4 technical replicates is presented. (**h**) Extract concentrations (mol/L) of selected intermediates of central carbon metabolism and (**i**) AECs as measured in *E. coli* extracted in ACN:water (1:1, *v/v*%) or in ACN:water:methanol (3:5:2, *v/v*%). The average and spread of n = 4 technical replicates are presented. Significance levels from two-tailed T-tests assuming equal variances are marked **: *p* ≤ 0.01; ***: *p* ≤ 0.001, in (**e**,**f**,**i**). Metabolite abbreviations are listed in Appendix A.

**Figure 6 metabolites-13-00150-f006:**
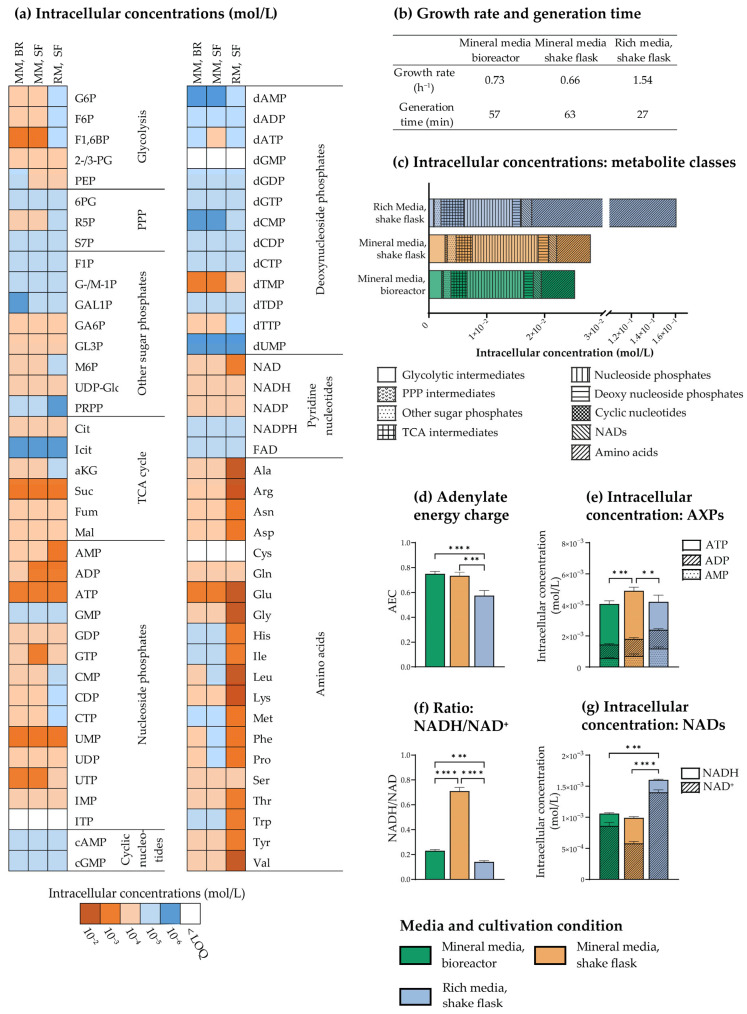
Metabolic profiling of mid-exponential phase *E. coli* (OD_600_ = 1) cultivated in shake flasks (SF) in rich (RM) or mineral medium (MM), or in benchtop bioreactors (BR) in MM. (**a**) Heat map of the average order of magnitude of intracellular concentrations (mol/L) in the three cultivation conditions. (**b**) Intracellular concentrations (mol/L) of different metabolite classes and pathway intermediates. Note that the abundances of pentose phosphate pathway (PPP) intermediates and cyclic nucleotides are too low to show in the figure. (**c**) Growth rate (h^−1^) and generation time (min) for a representative experiment. (**d**) Adenylate energy charge (AEC), (**e**) intracellular concentrations (mol/L) of AMP, ADP, and ATP, (**f**) NADH/NAD^+^ ratios, and (**g**) intracellular concentrations (mol/L) of NADH and NAD^+^ presented for all three cultivation conditions. The averages from n = 4 technical replicas from 1 (rich medium) or 3 (mineral media) biological replicates are presented in (**a**,**c**–**g**). Significance levels from a one-way ANOVA with Tukey’s multiple comparisons test are marked **: *p* ≤ 0.01; ***: *p* ≤ 0.001; ****: *p* ≤ 0.0001 in figure (**d**–**g**) (calculated for the total AXP and NAD pool in figure (**e**,**g**), respectively). TCA; tricarboxylic acid. Metabolite abbreviations are listed in Appendix A. Absolute intracellular concentrations are tabulated in Appendix A.

**Figure 7 metabolites-13-00150-f007:**
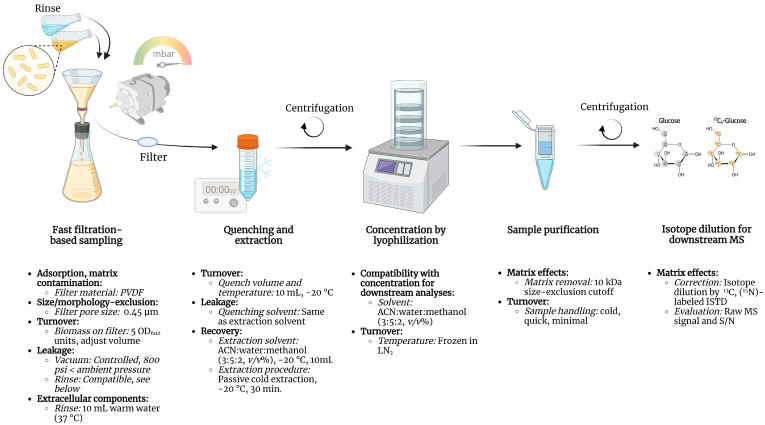
Simplified schematic of the optimized fast filtration sampling, quenching, extraction, and sample purification protocol for accurate, high-precision metabolic profiling of *E. coli*. Critical considerations (bold), parameters (italic), and recommended values, materials, and procedures discussed throughout this study are summarized in the lower panel. PVDF, polyvinylidene fluoride; ACN, acetonitrile; LN_2_, liquid nitrogen; ISTD, internal standard; MS, mass spectrometry; S/N, signal to noise.

**Table 1 metabolites-13-00150-t001:** Brand name, filter material, pore size (µm), supplier, and product number of 47 mm membrane disc filters tested for fast filtration-based sampling of *E. coli* for metabolite profiling. PVDF, polyvinylidene difluoride; PTFE, polytetrafluoroethylene.

Brand Name	Filter Material	Pore Size (µm)	Supplier	Product Number
Durapore	PVDF	0.22	Sigma-Aldrich	GVWP04700
Durapore	PVDF	0.45	Sigma-Aldrich	HVLP04700
Durapore	PVDF	0.65	Sigma-Aldrich	DVPP04700
Magna Nylon Filter	Nylon	0.22	GVS Life Sciences	1213769
Nylon Net Filter	Nylon	0.45	Sigma-Aldrich	HNWP04700
Omnipore	Hydrophilic PTFE	0.20	Sigma-Aldrich	JGWP04700
Omnipore	Hydrophilic PTFE	0.45	Sigma-Aldrich	JHWP04700

## Data Availability

The intracellular metabolite concentration data presented in this study are available in Appendix A.

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
