# Peer review of "Optimized Fast Filtration-Based Sampling and Extraction Enables Precise and Absolute Quantification of the Escherichia coli Central Carbon Metabolome"

_metabolites, 2023, doi:10.3390/metabo13020150_

Round 1

Reviewer 1 Report

In my opinion, the presented article is extremely relevant, and interesting and provides a detailed picture of the subject investigated.

The article thoroughly presents the topic, taking a look at various methodological aspects related to characterizing the central carbon metabolome of E. coli. Moreover, the text is well written, the structure makes the reading very understandable and smooth. The Results and Discussion are appropriate and worth publishing.

Reviewer 2 Report

The authors present a fast filtration-based sampling protocol for this widely applied model organism, focusing on pitfalls such as inefficient filtration, selective loss of biomass, matrix contamination, and membrane permeabilization and leakage, and achieve and demonstrate high precision and low metabolite turnover, and, followingly, minimal perturbation of the inherent metabolic state. These results were helpful for future central carbon metabolome in Escherichia coli. Below are some more detailed comments:

1. The format of the manuscript needs to be carefully revised, for example the “E. coli “in line 205, “h-1” in Fig6b, “13C” in line 852, and also please check the reference style to make it consistent.

2. What are the differences between this manuscript with the previously reports?

3. Is this possible to perform the central carbon metabolome analysis using the above method for a normal target products? And what’s the key point?

Reviewer 3 Report

In the reviewed manuscript, Thorfinnsdottir et al. present the development of an optimized fast-filtration and extraction method for the targeted absolute quantification of central carbon metabolites in E. coli. Finally, the optimized procedure is applied to three conditions, comparing and briefly discussing the different intracellular metabolite concentrations among the experimental groups. The manuscript is generally of excellent quality, and the data are presented professionally and scientifically sound. Although the topic has been addressed by other working groups in the past and the novelty can not be considered very high, it might still be of interest to a broad readership of the journal since the final method represents a relatively simple procedure and possible option for other scientists working in this area. Furthermore, the method might be reproduced without self-made rapid sampling or filtration devices and relies on standard laboratory equipment. Nonetheless, several points should be addressed before the article might be accepted for publication (minor revision). In this context, the comments below are expected to be helpful and contribute to further improving the quality of the manuscript.

General comments:

1. In general, it would be much appreciated if the authors would add a short paragraph addressing the overall advantage and uniqueness of their developed protocol compared to other approaches (for example, doi: 10.1007/s11306-014-0686-2). 

Specific comments:

1. Lines 47-49: It is said that “The choice of strategy depends both on the properties of the model species [14,15] and the culture conditions, and on the physicochemical properties of metabolites of interest“. This is correct – however, the available analytical method also plays a vital role in this context and might impose certain boundaries on the sampling and extraction procedure. Please complement the sentence accordingly.

2. Lines 88-101: Here, the scope and goal of the work are described. Especially the development of a reliable fast-filtration protocol for subsequent metabolomic measurements for E. coli has been the subject of several other studies (for example, doi: 10.1007/s11306-014-0686-2). Here, also in relation to general comment no.1, it would be much appreciated if the present work's novelty and impact would be highlighted more specifically. 

3. Lines 118-120: It is argued that the metabolite leakage from E. coli occurs unspecifically, resulting in lower AECs. Presumed the metabolite leakage would affect ATP, ADP and AMP unspecifically, then the AEC would not change (for example, check the discussion in doi: 10.1007/s11306-008-0114-6). In fact, the lower AEC might result from specific leakage and/or fast turnover of ATP upon leakage. Please comment and rephrase.

4. Line 187: Obviously, the developed fast-filtration method cannot be applied at high cell densities. However, is it possible from calculations or practical experience to say something about the maximal OD at which application of the developed method would make sense at the applied filtration pressure? When is filter rinsing no longer possible, or at which OD do other problems occur? This information would be helpful to readers who would like to apply this method, and the addition of this information would be much appreciated. 

5. Lines 428-432: Here, the importance of absolute metabolomic data (in mol/L) is pointed out. It is said that they can be related to biochemical parameters, such as Km values. This is true, but another decisive benefit of intracellular metabolite concentrations is that they can be exploited for mathematical frameworks, such as Metabolic Control Analysis or thermodynamic calculations. Please rephrase and/or complement.

6. For all subheadings: Check the formatting of the species names.

7. Lines 444-447: It is said that “By comparing the metabolite profiles of E. coli cultured on different carbon sources and under different levels of oxygenation presented herein, we could explore and add to the understanding of how these important parameters affect central carbon metabolism.“ In the following, the results for the three cultivations are presented and discussed. The exact composition of the two different media is stated in the Materials and methods section, and glucose (“mineral medium”) and yeast extract (“rich medium”) can be identified as potential carbon sources. Here, one might wonder what the main carbon compound in the yeast extract is and if it is appropriate in this context to present YE and glucose as two entirely different carbon sources. Is it possible that the YE also contains a significant number of carbohydrates, which are converted into glucose extra- or intracellularly and that the observed differences in metabolite concentrations do not mainly originate from the different carbon sources but simply from the differences in other media constituents? Please comment and change or weaken the statements accordingly.

8. Regarding figure 6: Please add information to the Materials and methods section on how the specific growth rate µ for the three cultivations was obtained. Simply from two data points or by regression of several points during exponential growth? It would also be much appreciated if the corresponding growth curves would be added as supplementary.

9. Figure 6 and section 2.5 in general: It is argued that the obtained data represent a comparison of E. coli’s metabolite profile on different carbon sources (see specific comment no. 7 in this regard) and different oxygenation levels. It can be assumed that the latter refers to the comparison of the cultivation on mineral media in the bioreactor (high DO level) and shake flasks (low DO level). The samples were taken at an OD of 1. Can it be assumed that, even at this low cell density, the oxygenation level in the shake flasks is already insufficient to meet the cell’s oxygen demand? If so, was the production of acetate observed in this case? Please comment.

10. Regarding the NADH/NAD ratio: NADH and NAD are well-known subjects to signal suppression and other issues in some MS-based metabolomic studies. It is assumed that those effects are mostly matrix-induced, but the exact cause is often unknown (see for instance, doi: 10.1186/s13068-017-0960-4). Might this have played a role in the significant differences in the observed NADH/NAD ratios in the present study? Why was the NADH/NAD couple not quantified additionally by commercially available kits to check if the data obtained from the MS quantification are plausible? Please comment.
